

# Technical note: Seamless extraction and analysis of river networks in R

Luca Carraro[1,2]

[1]Department of Evolutionary Biology and Environmental Studies, University of Zurich, Zürich, Switzerland
[2]Department of Aquatic Ecology, Swiss Federal Institute of Aquatic Science and Technology, Eawag, Dübendorf, Switzerland

**Correspondence:** Luca Carraro (luca.carraro@eawag.ch)

**Abstract.** Spatially explicit mathematical models are key to a mechanistic understanding of environmental processes in rivers. Such models necessitate extended information on networks' morphology, which is often retrieved from geographic information system (GIS) software, thus hindering the establishment of replicable, script-based workflows. Here I present `rivnet`, an R-package for GIS-free extraction and analysis of river networks based on digital elevation models (DEMs). The package exploits TauDEM's flow direction algorithm on user-provided or online accessible DEMs, and allows computing covariate values and assigning hydraulic variables across any network node. The package is designed so as to require minimal user input, while allowing customization for experienced users. It is specifically intended for application in models of ecohydrological, ecological or biogeochemical processes in rivers. As such, `rivnet` aims to make river network analysis accessible to users unfamiliar with GIS-based and geomorphological methods, and therefore enhance the use of spatially explicit models in rivers.

## 1 Introduction

Erosional forces exerted by surface waters shape the landscape, giving rise to runoff paths and drainage channels. The resulting river networks share universal features and scaling attributes (Rodríguez-Iturbe et al., 1992; Maritan et al., 1996; Rodríguez-Iturbe and Rinaldo, 2001; Carraro and Altermatt, 2022), which control a myriad of interrelated processes that ultimately characterize a riverine ecosystem, such as hydrological response (Marani et al., 1991; Blöschl and Sivapalan, 1995), substrate types (Frissell et al., 1986), sediment transport (Czuba, 2018), variability of habitat types and resulting biological communities (Vannote et al., 1980; Altermatt, 2013), dispersal pathways for organisms (Tonkin et al., 2018), and transport rates of organic and inorganic compounds (Jacquet et al., 2022; Yang et al., 2021a). As such, accurate landscape topography representation and characterization is a necessary component of riverine environmental modelling. Virtually any spatially explicit representation of processes taking place in a river network (whether it be a model of stream temperatures (Carraro et al., 2020c), species distribution (Buisson et al., 2008; Lois et al., 2015), epidemiological dynamics (Bertuzzo et al., 2010; Carraro et al., 2017), metabolic regimes (Segatto et al., 2021, 2023), microplastics (Uzun et al., 2022) or environmental DNA transport (Carraro et al., 2018)) requires the determination of flow paths, slopes, along-stream distances, upstream areas and connectivity patterns.

Digital elevation models (DEMs) provide gridded data on terrain elevation at different resolutions, and their analysis has become a cornerstone of geomorphological and hydrological studies (Kenward et al., 2000; Florinsky et al., 2002; Zhang



and Montgomery, 1994). Essentially, following established algorithms, flow directions are derived from the elevation map
(suitably modified in order to correct for local sinks) and subsequently transformed into drainage areas. Flow directions and
drainage areas then allow computing all other relevant properties (e.g., location of channel sources, Strahler stream order, reach
lengths) of a river network. Typically, DEM analyses are conducted on geographic information system (GIS) software, which
provides a graphical user interface and several features for spatial analyses, but generally lacks tools for subsequent advanced
modelling. Hence, terrain analysis resulting from GIS software must often be treated in other programming environments
(such as R, Python or MATLAB) in order to produce the desired output. The need for passing files back and forth between
GIS and programming software is not only cumbersome, but also hampers replicability, code sharing and adherence to best
practices in open research data. Furthermore, users with a more biology-oriented background might be unfamiliar with the
geomorphological methods used in GIS-based DEM analyses. Filling this gap is likely to promote the development of fruitful
cross-disciplinary studies.

Approaches for watershed delineation have been recently reviewed in David et al. (2023). Arguably the most popular toolset
for DEM analysis is TauDEM (Tarboton et al., 1991; Tarboton, 1997), which can be run both via an ArcGIS toolbox and
command line executables. TauDEM implements two main algorithms for evaluating drainage directions: single flow direction
(D8) (O'Callaghan and Mark, 1984) and multiple flow direction (D-infinity) (Tarboton, 1997). The D8 algorithm prescribes
that each cell drains into the lowest of its eight neighboring cells, whereas in D-infinity the steepest slope of a triangular
facet is used, and flow is partitioned into the two neighboring cells that are closest to the steepest flow direction. D8 has the
advantage of attributing univocal along-stream distances between two flow-connected points, but might tend to produce overly
concentrated drainage patterns. On the contrary, D-infinity allows for divergent flow, but at the cost of not defining along-stream
distances. More recent developments in watershed delineation algorithms focused on the use of hexagonal grid-based DEMs
(Wang and Ai, 2018), LiDAR data (Lyu et al., 2021) and flow enforcement (Wu et al., 2019) in order to enhance the accuracy
of the extracted river networks and the computational performance. An attempt at improving user experience is represented by
the ArcGIS USUAL toolbox (David et al., 2023) which allows simplified, automated workflows for watershed delineation and
is able to distinguish between tributary subcatchments and interfluvial areas.

A commonly used non GIS-based alternative for watershed delineation is TopoToolbox (Schwanghart and Kuhn, 2010),
which implements both single and multiple flow direction algorithms, but is developed for a proprietary programming environ-
ment (MATLAB). In the open source R language, the `dynatopGIS` package processes digital elevation data for subsequent
use in a semi-distributed hydrological model (TOPMODEL; Quinn et al. (1991); Beven and Freer (2001)), but requires both a
DEM and a river network shapefile as inputs. The `openSTARS` R package (Kattwinkel et al., 2020) delineates river networks
by interfacing with the open-source software GRASS GIS, and its scope is limited to the creation of spatial stream network
(`SSN`) objects for use in a class of spatial statistical models tailored for rivers (Ver Hoef et al., 2014). Other resources for
river network analysis in R do not use DEMs but river shapefiles and `igraph` objects (Csardi and Nepusz, 2006), and focus
on evaluating along-stream paths and distances (packages `riverdist` (Tyers, 2022) and `rivernet` (Reichert, 2020)), or
calculating connectivity and fragmentation indices (`riverconn` package; Baldan et al. (2022)).



Despite the long story of developments in digital elevation data and river network analysis, a DEM-based tool in the R
language for stream network delineation has been hitherto lacking. To fill this gap, I here present `rivnet`, a package allowing
seamless extraction of river networks from user-provided or open-source DEMs (e.g., Mapzen: https://registry.opendata.aws/
terrain-tiles/) and calculation of relevant properties for subsequent modelling purposes. It allows extracting river networks from
any region of the world and at a wide range of scales, that is from small-scale watersheds covering a few square kilometers up to
continental drainage basins. In particular, it enables extracting river network information independently of any other resources
(such as maps, political boundaries etc.) other than DEM, and is thus especially suitable for data-scarce regions. While global
datasets on river networks have recently been made available (Amatulli et al., 2022; Domisch et al., 2015), their resolutions
might be too coarse for studies based on small-scale basins, and the information contained therein might not fit all purposes.

The `rivnet` package is explicitly oriented towards modelling applications in ecohydrology, ecology and biogeosciences in
general. As such, the workflow for river network delineation is simplified and prearranged, which facilitates the task of users
unfamiliar with geomorphological tools. The choice of the R environment is also convenient because R is one of the most
widely used programming tools in the biological sciences. Building on open-source DEMs and R packages for their automatic
download (`elevatr`; Hollister et al. 2020), `rivnet` makes it possible to simply "point a finger on a map" and get the
corresponding watershed. The produced river objects are compatible with the `OCNet` package (Carraro et al., 2020a), which
generates Optimal Channel Networks, i.e. virtual analogues of river networks that are commonly used across a wide range of
ecological studies. As such, `river` objects generated by `rivnet` can be displayed and analyzed with `OCNet` functions, as
well as transformed into `igraph` or `SSN` objects for portability with other river-related R packages (Baldan et al., 2022; Csardi
and Nepusz, 2006; Ver Hoef et al., 2014). In the following, I discuss the design and main features of `rivnet`, and showcase
its use in two applications, the first being more geomorphology-oriented (evaluation of scaling features of river networks) and
the second implementing a simple biogeochemical model (release and transport of phosphorous at a catchment scale).

## 2  Package overview

In `rivnet`, river network extraction is performed via TauDEM's D8 flow algorithm, which allows univocal determination of
flow paths and along-stream distances. TauDEM command line executables are invoked via the `traudem` R package (Carraro
et al., 2022), which provides an interface to the TauDEM methods and guides the installation of the TauDEM C++ library,
as well as its dependencies GDAL and MPI, depending on the operating system used. The main function of the `rivnet`
package, `extract_river`, executes a fixed sequence of TauDEM commands to delineate the essential properties of a catch-
ment, that is flow directions and drainage areas (Fig. 1a). An optional parameter, `threshold`, can be passed to TauDEM's
`MoveOutletsToStream` to tune the subset of pixels (of drainage area larger than `threshold`) to which the outlet is to
be snapped, which thus enables a proper identification of the outlet location in the extracted stream network. In its simplest
application, `extract_river` only requires the coordinates of the desired outlet(s) and an user-provided DEM or, alter-
natively, the extent of the region from which digital elevation data are to be downloaded, together with information on the



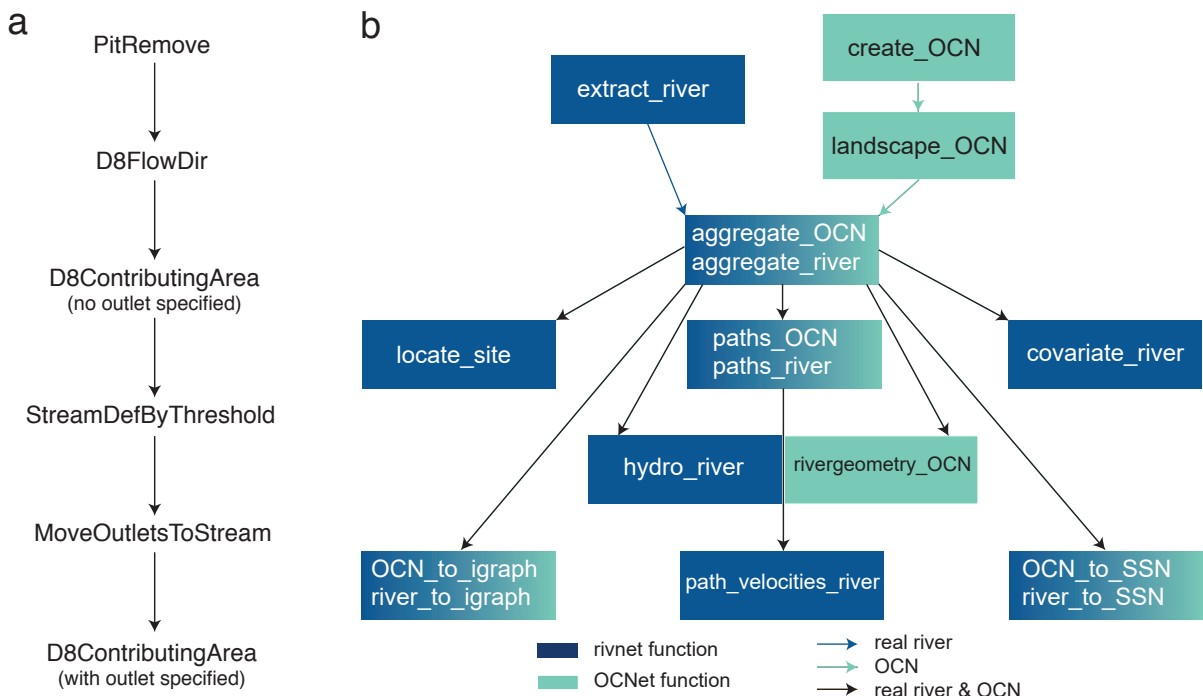

**Figure 1.** a) Sequence of TauDEM commands operated by `extract_river`. b) Structure of the `rivnet` and `OCNet` packages. In both panels, arrows indicate that the output of a function is used as input of the next function.

coordinate system, i.e. the EPSG identifier of the Geodetic Parameter Dataset. In the latter case, control is passed to function `get_elev_raster` of `elevatr` for downloading the DEM data, prior to execution of the TauDEM commands.

Outputs from `extract_river` include an optional plot of the drainage area map and contour of the catchment corresponding to the selected outlet (which can be used as a diagnostic tool to assess whether the input outlet coordinates lead
to the sought catchment; Fig. 2a), a raster file of pit-filled elevations, D8 flow directions and drainage areas (in which case `extract_river` serves as a one-line wrapper for the sequence of TauDEM commands of Fig. 1a), and an object of the newly defined `river` S4 class. Such an object is compatible for use in the `OCNet` package, according to the scheme of Fig. 1b; `river` objects are de facto lists containing sublists (hence elements of `river` objects can be accessed with both '$' and '@' operators), which in turn define the river network at different aggregation levels. Carraro et al. (2020a) provides an
extensive overview of definitions and use of aggregation levels, which are here briefly summarized for the sake of completeness. The flow direction (FD) level considers all DEM cells within the catchment as nodes of a network, and results from the application of `extract_river`. The so-obtained `river` object can be used as input into function `aggregate_river`, which mirrors `aggregate_OCN` of `OCNet`. This function defines three additional aggregation levels. The river network (RN) level is constituted by cells whose drainage area is larger than a given threshold value. A subset of such nodes then constitutes
the aggregated (AG) level: in the simplest setting, these are constituted by source and confluence nodes. Properties at the AG





level can either refer to the 'node' (i.e., source or confluence cells) or the 'edge' (i.e., the reach downstream of its associated node); for instance, reach length is provided for the edge, whereas drainage area values are provided with respect to both the node and the edge, in the latter case being equal to the drainage area at the downstream end of the reach. Additional nodes at the AG level can be included to split reaches that are excessively long (by imposing a maximum allowed reach length via option

`maxReachLength`) or in correspondence with point features that the user might want to include in their model or design (say, a dam or a wastewater treatment plant); for this latter purpose, option `breakpoints` can be used in `aggregate_river`. Finally, each so-obtained reach is associated to a subcatchment (SC level), that is the set of cells that directly drain into a reach.

Objects of the `river` class can be displayed using the `plot()` base function, which can call different drawing functions of `OCNet` depending on whether the object has been aggregated via `aggregate_river` and on optional themes to

be displayed along the river network (see an example in the Applications section). Aggregated `river` objects can be processed by `paths_river` (mirroring the analogous function from `OCNet`), which enables calculation of along-stream paths and distances between nodes at the RN and AG levels. Functions `river_to_igraph` and `river_to_SSN` mirror the corresponding functions of `OCNet` (Fig. 1b), and allow porting `river` objects in alternative formats.

Other functions have been specifically developed for the `rivnet` package, and are aimed to provide relevant variables for

use in ecohydrological and ecological applications. First, function `covariate_river` allows evaluating covariate values at the subcatchment (SC) level from user-provided raster files. Covariates are calculated with reference to both local (i.e., average values across a subcatchment) and upstream reference areas, and can be either of categorical (each cell of the input raster belongs to a different category, which is the case of e.g. land cover types) or continuous type (for instance, mean air temperature at each cell). Covariate values calculated by `covariate_river` are the fraction of cells covered by a specific

category within the reference area, or the mean value across cells within the reference area in the case of continuous inputs.

Second, function `hydro_river` provides a hydraulic geometry model for the whole river network based on well-established scaling relationships of hydraulic variables (Leopold and Maddock, 1953) and uniform flow relationships. The evaluated hydraulic variables (either at the RN or AG level) are river width, depth, water velocity, water volume, hydraulic radius and average bottom shear stress. The necessary input can be as minimal as one value of width and one value of depth or discharge

at two locations in the river network (which can, but do not have to be the same location for both measures). Depending on the type and number of width, discharge and depth values provided as input, `hydro_river` can calculate hydraulic variables in different ways (see the package documentation for details). When both discharge and depth data are provided to the function, preference for the use of power-law scaling relationships or uniform flow equation can be set by the user; if uniform flow equation is used, the roughness coefficient can be made spatially varying. The function can handle cross-sections of different

shapes, provided that they are vertically symmetric and that river width $w$ can be expressed as a power law of depth $d$. This formulation includes rectangular, triangular and natural sections (for which $w \sim d^{0.65}$, according to Leopold and Maddock 1953). While `hydro_model` assumes that the input hydraulic data are referred to the same time point (and hence the assignment of hydraulic variables across the network is a sort of spatial interpolation), the use of a non-rectangular cross section allows implementing variation of width over time, which for instance is of interest when studying the role of hydrologic contraction



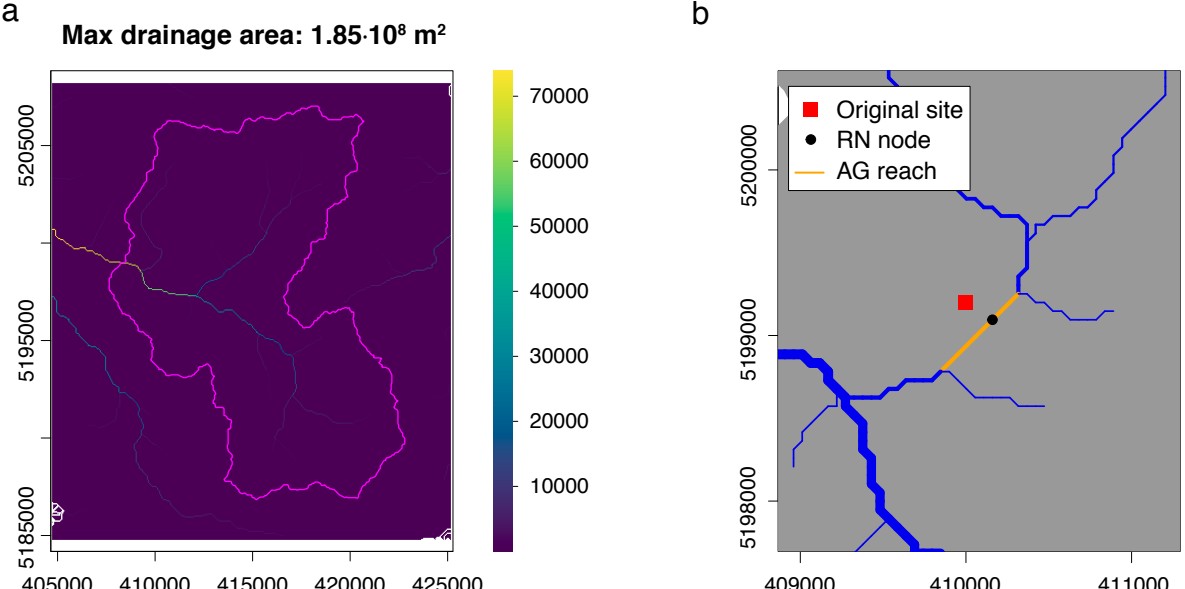

**Figure 2.** Examples of drawing options in `rivnet` functions. a) Use of option `showPlot = TRUE` in `extract_river` for the extraction of river Ilfis (see Fig. 3). The color map indicates the number of cells drained by each cell, while the contour of the catchment extracted is displayed in magenta. Coordinates are given in the projected reference system ED50/UTM 32N (identified by option `EPSG = 23032`). This can be used as a diagnostic tool in the case the extracted catchment is too small, or does not have the expected shape. In such cases, it is suggested to tweak the outlet coordinates, option `threshold`, or the DEM resolution. b) Use of option `showPlot = TRUE` in `locate_site` for the same river of panel a. A generic site (red square) is attributed to the closest RN and AG nodes (in this case, distance as the crow flies is minimized).

on organic matter dynamics (Catalàn et al., 2022). If temporal variation of hydraulic variables is required, `hydro_river` should be applied separately for each time point.

Third, function `path_velocities_river` can be applied after paths (via `paths_river`) and hydraulic variables (via `hydro_river`) have been computed in order to calculate mean water velocities across any path between two connected nodes (either at RN or AG level). These could be relevant for transport models, for instance in the case of environmental

DNA (Carraro et al., 2020b). Finally, `locate_site`, unlike the previous functions, does not produce a `river` object but identifies the RN and AG nodes that are closest to a given point, either as the crow flies or along the downstream direction. Coordinates of such point need to be inputted manually, but could be retrieved by clicking on a map via the `locator()` graphics function. Function `locate_site` also provides a drawing option, which plots a zoom-in of the `river` object in the area of interest, thus facilitating the correct identification of a site (Fig. 2b). This function allows pinpointing the location

of sampling sites across a river network, or other point features that might be included as nodes of the network (via the





`breakpoints` argument of `aggregate_river`), but it could also be used to evaluate the distance of non-riverine sites to the river network (e.g., meteorological stations or plot sampling sites for ecological assessments).

In the perspective of a joint use with `rivnet`, several functions of `OCNet` have been expanded and improved. In particular, as of `OCNet` v0.6.0, the speed of `aggregate_OCN` has increased substantially (about two orders of magnitude faster than
the previous version), `draw_subcatchments_OCN` allows displaying themes across subcatchments, the plotting algorithm in `draw_thematic_OCN` has been made more agile, and `OCN_to_SSN` allows input of observation and predicted design from user-specified coordinates.

## 3 Applications

### 3.1 Scaling properties of river networks

A first case study focuses on the evaluation of widely known elongation and aggregation properties of river networks. In particular, Hack's law (Hack, 1957) states that the length $L$ of the main river stem in a catchment scales as a power law of the drainage area $A$: $L \sim A^h$, where $h$ is termed Hack's exponent, and is expected to range between 0.45 and 0.7, with a value of 0.6 that is suitable for basins up to 20,000 km$^2$ (Sassolas-Serrayet et al., 2018). Within a catchment, the probability distribution of drainage areas is also expected to follow a power law: $P[A \geq a] \sim a^{-\beta}$, with $\beta = 0.43 \pm 0.02$ (Rinaldo et al.,
2014). The universality of interrelated power-law scaling relationships in drainage basins has been interpreted as the signature of the fractal character of river networks (Maritan et al., 1996). Performing such analysis in `rivnet` allows showing how areas, along-stream and out-of-stream distances, and different aggregation levels (RN, AG, SC) can be used.

Both Hack's law and the probability distribution of drainage areas were evaluated for four Swiss rivers of different total catchment area (range: 185–5329 km$^2$) extracted and analyzed via `rivnet`. Function `extract_river` was applied to a
Mapzen open-source DEM with cell resolution of about 50 m (the exact resolution depends on the latitude; see the documentation of `elevatr` for details). To evaluate Hack's law, the procedure described in Sassolas-Serrayet et al. (2018) was followed: a threshold area of 1 km$^2$ was used in `aggregate_river` to evaluate the network nodes at the AG level. Confluence nodes were used as sub-basin outlets where upstream areas $A$ and lengths $L$ were evaluated. Upstream lengths are made up of an along-stream component (evaluated via `paths_river`), i.e. the distance from the considered sub-basin outlet to a source
node, and an out-of-stream component evaluated as the horizontal distance from a source node to the subcatchment divide (Sassolas-Serrayet et al., 2018). The relationship between $L$ and $A$ for the four analyzed rivers is shown in Fig. 3a. Values of the Hack's exponent found for the four analyzed basins are within the expected range (Fig. 3a). The power-law scaling of drainage areas is also verified (Fig. 3b), with exponents that are in fair agreement with theoretical values (Rodríguez-Iturbe and Rinaldo, 2001).

This case study also allows an assessment of the computational performance of `rivnet`. As shown in Fig. 4, a relevant portion of the total computational time to produce the `river` objects used in this application is spent for DEM download and execution of TauDEM commands, while the total overhead required by functions `extract_river`, `aggregate_river` and `paths_river` is of the same order of magnitude as the two aforementioned tasks.





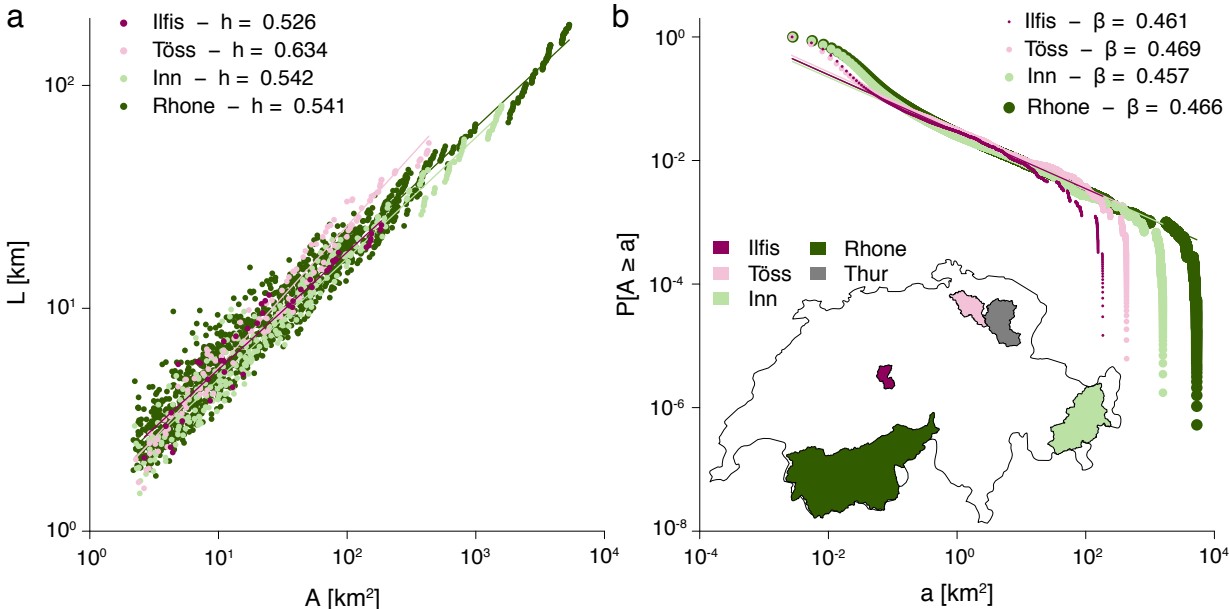

**Figure 3.** Results for the case study on scaling properties of river networks. a) Scaling of length of the main river stem $L$ with drainage area $A$ (Hack's law): $L \sim A^h$. Fitted Hack's exponent values for the four analyzed basins are reported in the legend. b) Probability distribution of drainage areas $P[A \geq a] \sim a^{-\beta}$. Fitted exponent values are reported in the legend. The inset displays the location of the four analyzed basins within Switzerland, as well as the Thur catchment (Fig. 5).

## 3.2 Release and transport of phosphorous at a catchment scale

The second case study applies a model of phosphorous dynamics to a Swiss river making use of `rivnet` functions. The model essentially follows Yang et al. (2021b), and it constitutes a component of the CⁿANDY (Coupled, Complex Algal-Nutrient DYnamics) model aimed at a parsimonious representation of dynamics of benthic and pelagic algae across a river network, and their interaction with a single limiting nutrient (phosphorous). In particular, this exercise illustrates the use of `rivnet` functions to calculate covariate values (`covariate_river`), assign hydraulic variables (`hydro_river`) and 190 pinpoint sites on the river network (`locate_site`).

According to Yang et al. (2021b), phosphorous inputs to the stream are related to both diffuse (i.e., depending on the different land use types) and point (i.e., related to wastewater treatment plants - WWTPs) sources, and are indicated with $\phi_{D,i}$ and $\phi_{P,i}$ [g P d$^{-1}$], respectively, where subscript $i$ indicates a given reach (node at the AG level) of the network. Phosphorous is assumed to be advected by streamflow while processed by the microbial community. The model reads:

$$\frac{\mathrm{d}P_i}{\mathrm{d}t} = \phi_{D,i} + \phi_{P,i} + \sum_{j=1}^{N} w_{ji} \frac{Q_j}{V_j} P_j - \frac{Q_i}{V_i} P_i - \frac{v_f}{d_i} P_i \qquad (1)$$

where $P_i$ [g P] is the phosphorous mass in reach $i$, $w_{ji}$ is an element of the adjacency matrix (equal to 1 if node $j$ drains into $i$, and null otherwise), $Q_i$ [m$^3$s$^{-1}$], $V_i$ [m$^3$] and $d_i$ [m] the water discharge, water volume and river depth in reach $i$, respectively;



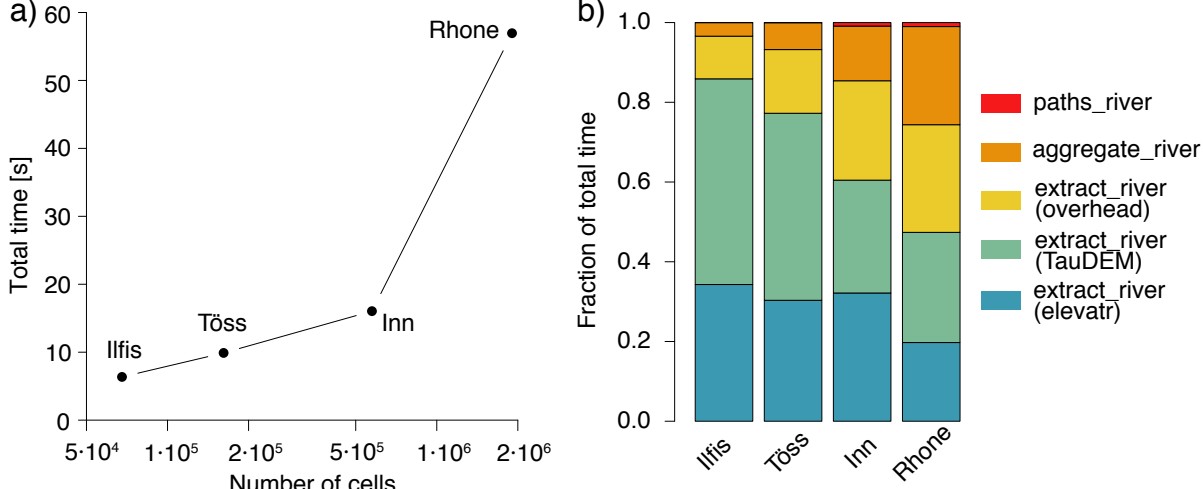

**Figure 4.** Estimates of computational time needed to produce the `river` objects used in the analysis of Fig. 3. a) Total time required as a function of the number of cells in the catchment. b) Breakdown of time for the different `rivnet` functions used: in particular, the time spent by `extract_river` is partitioned into time required by DEM downloading ("elevatr"), application of TauDEM commands ("TauDEM"), and creation of a `river` object ("overhead"). Results are the average of five runs operated via an Intel® quad core i7-7700K processor with 16 GB RAM; TauDEM commands were run with 8 parallel processors.

and $v_f$ [ms$^{-1}$] is an uptake velocity accounting for transformation processes operated by microbial biofilm in the hyporheic zone (Basu et al., 2011). Point phosphorous loads were estimated as $\phi_{P,i} = k_P H_i$, where $k_P$ [g P day$^{-1}$] is the phosphorous load per equivalent inhabitant, and $H_i$ the population equivalent of the WWTP located in reach $i$. Diffuse loads were estimated

based on three main land cover types (forested, agricultural and urban areas) as $\phi_{D,i} = (k_F f_{F,i} + k_A f_{A,i} + k_U f_{U,i})A_{S,i}$, where $A_{S,i}$ [m$^2$] is the area of the subcatchment pertaining to reach $i$; $f_{F,i}$, $f_{A,i}$ and $f_{U,i}$ the fractions of $A_{S,i}$ covered by forested, agricultural and urban areas, respectively; and $k_F$, $k_A$ and $k_U$ [mg P m$^{-2}$day$^{-1}$] their respective phosphorous load per unit area.

Model Eq. (1) was applied to evaluate steady-state phosphorous concentrations in a Swiss catchment (the Thur, Figs. 3b
and 5a), which is known to have very different environmental conditions among its main tributaries (Carraro et al., 2020b). The river network was extracted via `extract_river` from the same DEM as in the previous application, and was aggregated (via `aggregate_river`) to a total of 413 nodes at the AG level by imposing a threshold area of 1 km$^2$ and a maximum reach length of 2500 m. Data on WWTPs and a raster land cover map for Switzerland were retrieved from open-source databases of the Swiss Federal Office for the Environment (FOEN). A total of 21 WWTPs are present in the Thur catchment (Fig. 5b); their
attribution to the AG nodes of the river network was performed via `locate_site`, while `covariate_river` was used to evaluate $f_{F,i}$, $f_{A,i}$ and $f_{U,i}$. Four hydrological stations operated by FOEN are located within the catchment (Fig. 5a); their corresponding AG nodes were assessed via `locate_site`. At these locations, river width was estimated via aerial images, and mean water discharges in the period 2012-2021 were calculated from the FOEN data. Function `hydro_river` was used



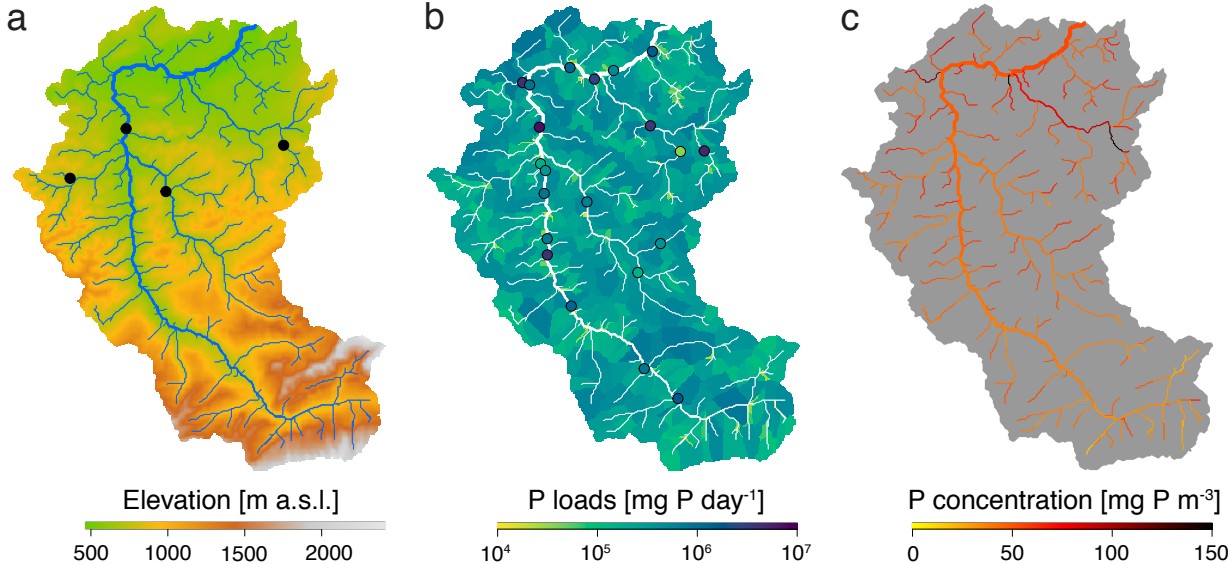

**Figure 5.** Results for the case study on phosphorous release and transport. a) Elevation map of the Thur catchment. Black dots indicate the locations of the hydrological stations. b) Map of phosphorous loads: diffuse loads $\phi_{D,i}$ are color-coded on the respective subcatchments, while point loads $\phi_{P,i}$ are displayed as dots. c) Map of phosphorous concentration as predicted by the model Eq. (1). The three panels were obtained as specific cases of the `plot()` function.

with such inputs to assign hydraulic variables across the network, by assuming uniform flow conditions and a spatially constant

Gauckler-Strickler roughness coefficient equal to 30 m$^{1/3}$s$^{-1}$, and natural river cross-sections. Model parameters were taken as in Yang et al. (2021b): $v_f = 0.17$ m day$^{-1}$, $k_P = 150$ mg P day$^{-1}$ (per inhabitant), $k_F = 0.1$ mg P m$^{-2}$day$^{-1}$, $k_A = 0.15$ mg P m$^{-2}$ day$^{-1}$, $k_U = 0.3$ mg P m$^{-2}$ day$^{-1}$. Phosphorous concentrations predicted by Eq. (1) are shown in Fig. 5c. Highest phosphorous loads are observed in the north-eastern part of the Thur catchment, in correspondence with a large WWTP located in a small reach (Fig. 5b).

**4  Perspectives and limitations**

The `rivnet` package enables the analysis of river networks in the R environment with a particular emphasis on environmental modelling purposes. Being based on digital elevation data (rather than river shapefiles), it allows computation of areas and thus attribution of covariate values or hydraulic variables across the river network. Extraction of drainage patterns is based on TauDEM's D8 flow direction algorithm, a robust and widely used method that allows univocal evaluation of distances between

stream sites. A major asset of `rivnet` is that, unlike other DEM-based river network analysis tools, it does not require the use of a graphical user interface-based GIS software, and hence it allows replicable workflows, while at the same time it provides a series of drawing options and functions assisting tasks such as delineating the catchment shape or localizing a site within

the river. The use of a fixed sequence of TauDEM commands is advantageous, because it makes the task of extracting a river network accessible to users unfamiliar with geomorphologic, hydrologic or GIS tools. In the same spirit, `rivnet` functions are designed so as to require the minimum amount of user input: by interfacing with the `elevatr` package, it is sufficient to provide the extent of the region of interest and outlet coordinates in order to obtain river data, and minimal information on river hydraulics (say, one width and one discharge value) is enough to produce a hydraulic model for the whole river network. At the same time, a wide set of options warrants customization for the more experienced users.

The main limitations of the `rivnet` package are inherent to the use of the D8 method. In particular, being based on DEM analysis, the accuracy in the determination of flow directions is constrained by the quality of the topographic data provided. This implies that river network delineation in low-relief areas could be imprecise, as algorithms specifically tailored for flat landscapes (e.g. Passalacqua et al., 2012) are not included in the current workflow. To help overcome such limitation, as well as those induced by artificial flow paths (such as ditches or pipes across elevation boundaries), it is thus suggested to manually decrease the elevation of the affected cells in the input DEM. Additionally, the use of a single flow direction algorithm such as D8 does not allow handling braided channels. In fact, TauDEM's D8 method, and in turn the `rivnet` package, is aimed towards catchment-scale terrain analysis, and hence topographic features at a reach scale might not be accurately reproduced.

In conclusion, because of its simplicity and wide range of possible applications, `rivnet` can facilitate and promote the development of spatially explicit ecological and biogeochemical models in river networks, which are crucial tools for a mechanistic understanding of the state and change of freshwater environments.

*Code and data availability.* The `rivnet` package is available on CRAN at the following link: https://cran.r-project.org/web/packages/rivnet/index.html. Scripts reproducing the applications are available at the following link: https://github.com/lucarraro/test_rivnet. Data used in the applications are freely accesible online (links are provided in the aforementioned scripts).

*Author contributions.* The author conceptualized the study, developed the R package, designed the applications and wrote the manuscript.

*Competing interests.* The author declares no competing interests.

*Acknowledgements.* Funding is from the Swiss National Science Foundation Ambizione grant PZ00P2_202010. The author thanks Florian Altermatt, Enrico Bertuzzo and François Keck for their helpful comments and feedback on the package and manuscript.



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
