# Peer review of "Technical note: Seamless extraction and analysis of river networks in R"

_EGUsphere, 2023_

## Referee Comment (RC1)

Review of the manuscript '*Technical note: Seamless extraction and analysis of river networks in R*' by *Carraro* submitted to *HESS*.

Recommendation: ACCEPT.

Focus of the paper: the author presented a novel package "rivnet", aimed at hydrological analysis and modelling river networks using digital elevation models (DEMs). The author developed his own novel package 'rivnet'. Therefore, the paper presents a valuable contribution to the scientific methods development in Geosciences with a special focus on hydrological modelling.

Relevance: The presented study is the original primary research within scope of the journal *Hydrology and Earth System Sciences.* The author developed a new package which focuses on modelling of the environmental processes in rivers. The manuscript meets general criteria of the significance in applied programming (R), environmental data modelling and hydrology (analysis of catchment, watershed basin analysis, etc). It is relevant to the journal topic as corresponding to the major domain and research disciplines: *hydrological modelling, Earth sciences, applied programming.*

Abstract is well written, concisely and clearly describes the undertaken study.

Structure: The article is well organized with 4 structured sections. 1 Introduction (a background on hydrological modelling is provided); 2 Package overview (here, the author presents the package in details and well explains technical nuances); 3 Applications (two case studies); 4 Perspectives and limitations.

Logic: The clarity of the text logic and organization of the paper is sufficient with details on technical description of the package and theoretical basis of hydrological modelling. The paper consistently interprets the results with detailed explanations and comments on new package 'rivnet'. A brief comparison of the package with those in previous approaches in hydrological modelling is presented.

Introduction presents a background, defines research goals and provides a clear statement of research problem. The Introduction well describes the research. Introduction and background show context of the article. Literature is well referenced and relevant.

Study area: is described with sufficient details.

Research questions and goal are identified: the author developed a novel package based on R syntax which aims at mathematical modelling of the hydrological processes. Objectives are relevant to the study aim with regard to river modelling (two cases of river modelling in study area of Switzerland).

Literature regarding the relevant topics is reviewed, formatted according to the journal rules and appropriately referenced. Evaluation diverse hydrological modelling aspects. Examples include papers on flow directions, upslope areas in grid DEMs, modelling river basins by fractals, GIS, modelling watersheds, geomorphic feature extraction from LiDAR etc. The literature is relevant and well selected.

Research gaps and weakness in former works are described lack of script-based hydrological package which would be used instead of GIS with extended functionality using R language.

Motivation is explained: this study contributes to fill in the gaps in the existing methods on river networks' morphology modelling using replicable, script-based workflows. The functionality of the presented package 'rivnet' is well explained with provided details on commands.

English language: fine.

Data used in this study are described: To illustrate the functionality of package, the author presented two cases: 1) evaluation of elongation and aggregation properties of river networks based on data on main river stem in a catchment scales. This well demonstrated the computational performance of 'rivnet' package; 2) modelling phosphorous dynamics based on data regarding selected Swiss rivers; Here the author demonstrated how the package can calculate covariate values, assign hydraulic variables and pinpoint sites on the river network. Both case studies are well explained and illustrated.

Methods: Methods described with sufficient detail and information. The author explained the main functionality of the developed R package: it exploits the DEM's flow direction algorithm based on

DEMs, computes covariate values and assigns hydraulic variables across network nodes. The workflow is well structured and clearly described with sufficient information to reproduce the approach.

Results are reported: The main results consists in the novel developed package - the 'rivnet'. The author well explained the functionality, demonstrated its syntax and presented 2 case studies. The package enables the analysis of river networks in the R language. The emphasis is placed on environmental modelling and hydrological data analysis (and DEM). The package allows computation of areas and compute covariate values or hydraulic variables across the river network. Extraction of drainage patterns is based on TauDEM's D8 flow direction algorithm. The author well explained and commented on the new developed package 'rivnet'.

Discussion interpreted the major outcomes of this study. The advantages of the obtained results are described and compared with other studies. The Discussion described the issues of methodology and results.

Conclusion Conclusions are well stated: the 'rivnet' aims to make river network analysis supported by geomorphological modelling and the use of DEM. The Conclusions are linked to original research question, explain and support the results (developed package) and summarized the study. The conclusions are appropriately stated and connected to the original research goal of development of the hydrological package.

Actuality, novelty and importance of the research is clear. It consists in the novel developed package based on R which aims at hydrological modelling. The package is flexible enough: it requires minimal user input, and allows customization for experienced users.

Academic contribution: Rigorous investigation performed to a high technical and professional standard in software development in geosciences. The author presented a package which is intended for application in models of ecohydrological, ecological or biogeochemical processes in rivers. In this way, the novel package is multi-purpose and has potentially diverse applications. The new R package "rivnet" will support hydrological analysis, spatial modelling and investigation of geomorphic and environmental processes. Such study well deserved to be published in HESS.

Figures The authors presented 5 figures which well illustrate the functionality of the novel package. The figures are of fine quality, the illustrations are well done with carefully selected type of lines and colour palettes, aesthetically plotted and professionally made from the cartographic perspective. The figures are easy to read for hydrologists and geographers, relevant and suitable to this paper. Figures are labelled and appropriately described. They illustrate the applications of the new 'rivnet' package.

Recommendation: This manuscript can be ==ACCEPTED== based on the detailed report above.

With kind regards,

- Polina Lemenkova.

24.05.2023.

---

## Author Response (AR1)

**Response to reviewers**

**Reviewer 1**

Thank you very much for your very positive assessment of this work. As it seems, no replies to specific comments or manuscript revisions are requested.

**Reviewer 2**

Thank you very much for your very positive assessment of this work. Specific comments are hereafter addressed.

**Fig. 1**. I agree that the distinction between arrow types in Fig. 1b may be unclear. In the revised manuscript, I used dashed arrows for category "real river & OCN", hence making the distinction clearer. Thanks for spotting this. Moreover, I modified Fig. 1b by adding a box for function "river_to_AEM", which has been added to the package as of version 0.3.0. An explanation of this function's scope has been added to the text.

**Fig. 2**. The river structure in Fig. 2a is indeed hardly visible, which is due to each pixel being assigned a color, as opposed to the catchment outline, which is a polygon and can thus be plotted with increased line width. This is unfortunately unavoidable, because this figure appears when calling extract_river(), that is before streams have been delimited (which actually occurs when the subsequent function aggregate_river() is called). I already tried alternative color palettes, which however did not improve the quality of the figure. In the revised manuscript, In the revised manuscript, I recreated Fig. 2a with a coarser DEM resolution, thus resulting in larger pixels and hence better readability of the figure. By the way, the catchment shown in Fig. 2a is river Ilfis, as already mentioned in the caption.

**Fig. 3b**. Thank you for this suggestion. However, I do not think that the inset figure (i.e., the map of Switzerland) should be a figure in itself, as this does not show an application of the use of rivnet, but only supports the two applications shown. However, I agree that some more details could be added to the inset. In the revised manuscript, I placed the y-axis of Fig. 3b on the right-hand side, thus making space for a bigger inset, in which borders of neighbouring countries, and meridians and parallels have been added.

**Use of D8 in combination with Dinf**. Thank you very much for this insightful comment. It is indeed possible to extract catchment features obtained via the D8 method and apply them in combination with the Dinf method (or alternative algorithms) for more detailed catchment analyses. Indeed, the application of the Dinf method from the TauDEM library in R is allowed by the general-purpose function taudem_exec() of package traudem, while streamlines derived via the D8 method are produced as output of rivnet's aggregate_river(). Nonetheless, I deem that building a dedicated workflow for this option in rivnet would be out of scope with respect to the main focus of the package, which is supporting ecohydrological, ecological and biogeochemical models at catchment scale. However, in the revised manuscript, I mentioned in the concluding section that it is possible to export river objects in free format and use them in subsequent landscape analyses with the Dinf or other methods.